# Superior photo-carrier diffusion dynamics in organic-inorganic hybrid perovskites revealed by spatiotemporal conductivity imaging

Xuejian Ma [1,4], Fei Zhang [2,4], Zhaodong Chu[1], Ji Hao[2], Xihan Chen [2], Jiamin Quan[1], Zhiyuan Huang[2], Xiaoming Wang[3], Xiaoqin Li [1], Yanfa Yan [3], Kai Zhu [2✉] & Keji Lai [1✉]

The outstanding performance of organic-inorganic metal trihalide solar cells benefits from the exceptional photo-physical properties of both electrons and holes in the material. Here, we directly probe the free-carrier dynamics in Cs-doped $FAPbI_3$ thin films by spatiotemporal photoconductivity imaging. Using charge transport layers to selectively quench one type of carriers, we show that the two relaxation times on the order of $1 \, \mu s$ and $10 \, \mu s$ correspond to the lifetimes of electrons and holes in $FACsPbI_3$, respectively. Strikingly, the diffusion mapping indicates that the difference in electron/hole lifetimes is largely compensated by their disparate mobility. Consequently, the long diffusion lengths ($3{\sim}5 \, \mu m$) of both carriers are comparable to each other, a feature closely related to the unique charge trapping and de-trapping processes in hybrid trihalide perovskites. Our results unveil the origin of superior diffusion dynamics in this material, crucially important for solar-cell applications.

[1] Department of Physics, University of Texas at Austin, Austin, TX, USA. [2] Chemistry and Nanoscience Center, National Renewable Energy Laboratory, Golden, Colorado, CO, USA. [3] Department of Physics and Astronomy, University of Toledo, Toledo, OH, USA. [4] These authors contributed equally: Xuejian Ma, Fei Zhang. ✉email: kai.zhu@nrel.gov; kejilai@physics.utexas.edu

Organic-inorganic lead trihalide perovskite solar cells (PSCs) have been in the limelight of photovoltaic research[1–3], as exemplified by the outstanding certified power conversion efficiency (PCE) that exceeds 25% to date[4]. Even in the polycrystalline form, the PSC thin films exhibit many remarkable photo-physical properties, such as high absorption coefficient[5], long carrier lifetimes[6], and low impurity scattering rate[7,8]. For photovoltaic applications, a particularly attractive feature of hybrid perovskites is that both electrons and holes are active in the photoconduction process[6–10]. From the theoretical point of view, the two types of carriers are expected to exhibit comparable effective mass, intrinsic mobility, recombination lifetime, and diffusion length[11–13]. In real materials, however, the balance between electrons and holes is usually broken by thin-film deposition conditions, defect structures, ionic disorders, and other sample-dependent parameters[14–17], which may affect the photo-carrier extraction in functional devices. A thorough understanding of the dynamics of individual charge carriers is thus imperative for continuous improvements of PSC performance towards commercial applications.

The spatiotemporal dynamics of electrons and holes in optoelectronic materials are widely studied by optical measurements such as time-resolved photoluminescence (TRPL) and transient absorption spectroscopy (TAS)[8–10,18,19]. The diffusion length can then be deduced by fitting the results to a diffusion model[8–10]. It should be noted that TRPL and TAS probe optical excited states and are often dominated by transitions with large oscillator strength. In contrast, the transport of photoexcited carriers is electrical and quasi-static in nature. In order to directly evaluate photoconduction, it is common to measure the photocurrent across electrical contacts, such as scanning photocurrent microscopy (SPCM)[16,20,21]. The spatial resolution of SPCM is diffraction-limited and the temporal response is dominated by the carrier transit time and extrinsic metal-semiconductor Schottky effect. In recent years, noncontact methods such as time-resolved microwave conductivity (TRMC)[6,7,16,22–25] and time-resolved THz spectroscopy (TRTS)[17,26] are developed to probe the photo-carrier dynamics. These far-field techniques, however, do not offer spatially resolved information such as diffusion patterns.

In this article, we directly probe free-carrier diffusion dynamics in Cs-doped formamidinium (FA) lead trihalide (FACsPbI₃) thin films by laser-illuminated microwave impedance microscopy (iMIM), a unique optical-pump-electrical-probe technique with 20-nm spatial resolution and 10-ns temporal resolution for the electrical detection[27]. The steady-state iMIM experiment addresses the most important photo-physical process in solar cells, i.e., the transport of photo-generated mobile carriers under the continuous illumination of ~1 Sun. The time-resolved iMIM (tr-iMIM) experiment detects the free-carrier lifetime that is also highly relevant for photoconduction. By depositing a hole or electron transport layer (HTL/ETL) underneath FACsPbI₃, we show that the two decay constants in tr-iMIM measurements are associated with the lifetimes of electrons and holes. The spatiotemporal imaging allows us to determine the diffusion coefficients, steady-state carrier density, and mobility of individual carriers. Interestingly, while the lifetime and mobility of electrons and holes differ substantially, their products and thus diffusion lengths are comparable to each other, which is likely due to the unique defect structures and charge trapping events in PSC thin films. Our results highlight the origin of nearly balanced diffusion dynamics of electrons and holes in hybrid trihalide perovskites, which is highly desirable for photovoltaic applications.

## Results

The PSC thin film in this study, hereafter labeled as Sample A, is 5% Cs-doped FA lead triiodide deposited on cover glasses (see

Methods). Compared with methylammonium (MA) based perovskites, FAPbI₃ exhibits superior stability at elevated temperatures and an ideal band gap for sunlight absorption[28]. The Cs-doping further stabilizes the room-temperature photo-active α-phase by reducing the Goldschmidt tolerance factor[29–32]. Perovskite films were deposited using the typical anti-solvent-assisted spin-coating procedure. The samples were then capped by spin-coating 15 mg ml⁻¹ PMMA (Mw ~ 120,000) film in chlorobenzene solution. For iMIM measurements, we chose a film thickness of $d = 250$ nm that is greater than the absorption length, such that light is fully absorbed, but much less than the carrier diffusion length, such that the photoconductivity is uniformly distributed in the vertical direction within the relevant time scale in our experiment. External quantum efficiency (EQE) spectra were also measured (Supplementary Fig. 1), showing good photoresponse across the solar spectrum. PSC devices made from the same material but with thicker film (550 nm) demonstrated a PCE above 20% under the standard air mass (AM) 1.5 illumination (Supplementary Fig. 1).

The spatiotemporal iMIM setup with a focused laser beam illuminating from below the sample stage is illustrated in Fig. 1a. In the tip-scan mode, the laser is focused by one set of piezo-stage and the second piezo-scanner carries the tip to scan over the sample[27]. In the sample-scan mode, the first set of piezo-stage aligns the center of the laser spot to the tip, whereas the sample is set in motion by the piezo-scanner[33,34]. In both configurations, one can fix the relative position between tip and sample and perform time-resolved (tr-iMIM) measurements[27]. Here the laser output is modulated by an electro-optic modulator (EOM), which is driven by a 7-kHz square wave from a function generator such that steady-state photoconductivity is reached in the laser-ON state and zero photoconductivity in the laser-OFF state. The same waveform also triggers a high-speed oscilloscope for iMIM measurement. The temporal resolution of our setup is ~10 ns (see Methods). The microwave electronics detect the tip-sample impedance, from which the local conductivity can be deduced[35]. The optical excitation in our setup is diffraction-limited, whereas the electrical imaging has a spatial resolution of 20–50 nm compared to the tip diameter. Quantification of the iMIM signals by finite-element analysis (FEA)[36] is detailed in Supplementary Fig. 2.

Figure 1b shows the iMIM images when Sample A was illuminated by a 446-nm ($h\nu = 2.78$ eV) diode laser with the intensity at the center of the laser spot $P_C = 100$ mW/cm², i.e., on the order of 1 Sun. The granular features are due to topographic crosstalk with the polycrystalline sample surface[33]. It is nevertheless evident that the photoresponse is continuous across many grain boundaries (GBs). Based on the iMIM response (Supplementary Fig. 2), we can replot the data to a conductivity map (Fig. 1c) with high conversion fidelity. For comparison, the optical image of the laser spot acquired from a charge-coupled device (CCD) camera shows a much smaller spatial spread in Fig. 1d. To improve the signal-to-noise ratio and minimize the topographic artifact, we averaged eight line profiles shown in Fig. 1c. The resultant curve, plotted in Fig. 1e, is clearly broader than the Gaussian beam profile ($e^{-r^2/w^2}$, $w \sim 2$ μm). Assuming that the carrier mobility μ is independent of charge density $n$ within the range of our experiment, the measured photoconductivity profile $\sigma(r)$ is proportional to the steady-state density profile $n(r)$ as

$$\sigma(r) = n(r)q\mu \quad (1)$$

where $q$ is the elemental charge. Here $n(r)$ can be described by the diffusion equation[27,37,38]

$$n(r) - L^2\nabla^2 n(r) = \frac{\eta}{d}\frac{P_c\tau}{h\nu}e^{-r^2/w^2} \quad (2)$$

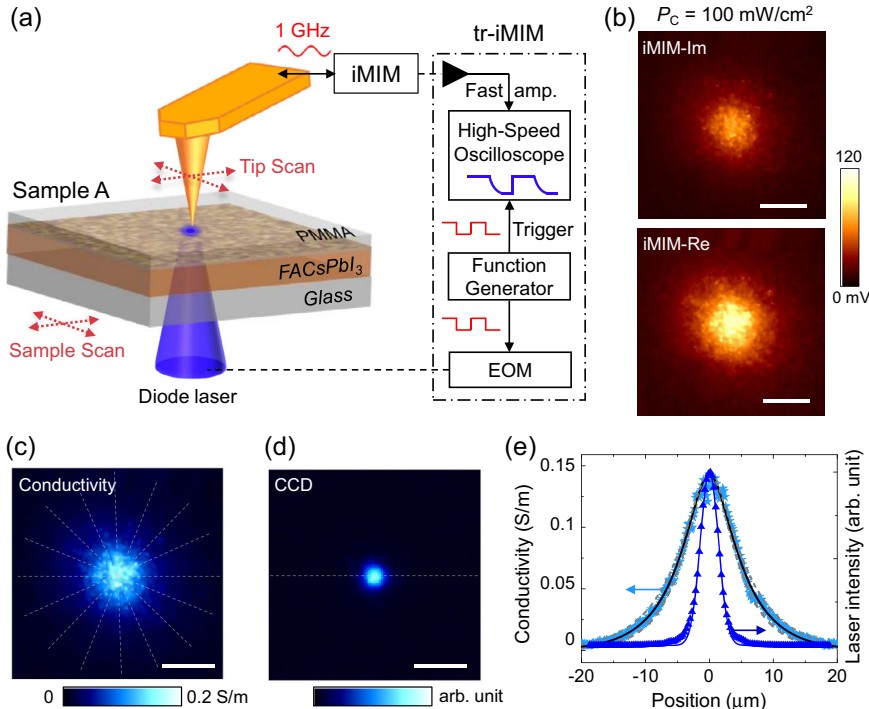

**Fig. 1 Photoconductivity mapping on FACsPbI₃ and diffusion analysis. a** Schematic of the iMIM setup with either the tip-scan or sample-scan mode. The tr-iMIM configuration is shown inside the dash-dotted box. The FACsPbI₃ thin film deposited on a glass substrate and encapsulated by a PMMA layer (Sample A) is also illustrated. **b** Tip-scan iMIM images when the sample is illuminated by a 446-nm diode laser at $P_C = 100$ mW/cm². **c** Photoconductivity map based on the iMIM data and FEA simulation. The dashed lines are various linecuts for the calculation of average signals. **d** Image of the laser spot taken by a CCD camera. **e** Line profiles of averaged photoconductivity and laser intensity, from which the diffusion length can be extracted. The solid black and dashed gray lines represent the best curve fitting and upper/lower bounds, respectively. All scale bars are 10 μm.

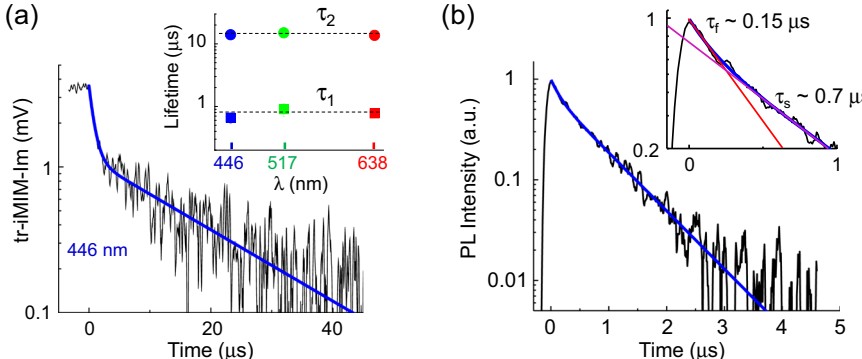

**Fig. 2 Time-resolved iMIM and carrier lifetime. a** Typical tr-iMIM relaxation curve of Sample A. The sample was illuminated by the 446-nm laser at $P_C = 100$ mW/cm² before $t = 0$ μs. The inset shows the two lifetimes under excitation lasers with wavelengths of 446, 517, and 638 nm. **b** TRPL data of Sample A. The blue curve is a biexponential fit to the TRPL data. The inset is a close-up view, showing the two decay time constants from the fitting.

where $L = \sqrt{D\tau}$ is the diffusion length, $D$ the diffusion coefficient, $\tau$ the lifetime, and $\eta \sim 1$ the incident photon-to-current conversion efficiency (IPCE). The analytical solution to Eq. (2) is

$$n(r) \propto \int_{-\infty}^{\infty} K_0(r'/L) e^{-(r-r')^2/w^2} dr' \quad (3)$$

where $K_0$ is the modified Bessel function of the second kind. By fitting the iMIM data to Eq. (3), we obtain a diffusion length $L = 5.1 \pm 0.6$ μm, consistent with values reported in the literature for thin-film PSCs[9,10,16,46]. As the laser power increases, $L$ decreases to ~4 μm at $P_C \sim 10^3$ mW/cm² and saturates at 3.5 μm for $P_C \sim 10^4$ mW/cm² (Supplementary Fig. 3).

Figure 2a shows a typical tr-iMIM decay curve (averaged over 242,880 cycles) on Sample A illuminated by the 446-nm laser at

$P_C = 100$ mW/cm² before $t = 0$. As the iMIM-Im signal scales with photoconductivity in our measurement range (Supplementary Fig. 4), we will just present the raw data in the following analysis. From Eq. (1), the decay of tr-iMIM-Im signal provides a direct measure of the lifetime of mobile carriers in the conduction or valence band. The relaxation fits nicely to a biexponential function $y = A_1 e^{-t/\tau_1} + A_2 e^{-t/\tau_2}$, with $\tau_1 \sim 0.7$ μs and $\tau_2 \sim 10$ μs. As shown in the inset of Fig. 2a, we observed the same $\tau_1$ and $\tau_2$ when using 517-nm and 638-nm lasers (Supplementary Fig. 5), suggesting that the time constants are intrinsic to the sample and independent on the laser wavelength. In contrast, the TRPL data on Sample A (Fig. 2b) exhibit two different times $\tau_f \sim 150$ ns and $\tau_s \sim 0.7$ μs, which will be discussed in the next section. By parking the tip at various locations of the film and measuring the decay

curves, we also show that the tr-iMIM response is spatially uniform on the sample surface within statistical errors (Supplementary Figs. 6 and 7).

In order to shed some light on the tr-iMIM data, we studied the carrier dynamics in perovskite thin films with electron or hole transport layers[6,7,9,10,22,23]. For the HTL sample, hereafter referred to as Sample B, a 20-nm PTAA (poly-triaryl amine) was spin-coated on the substrate before the same 250-nm FACsPbI$_3$/30-nm PMMA film was deposited. The PTAA layer rapidly extracts photo-generated holes from FACsPbI$_3$ within a sub-10-ns time scale[22,23,39]. Similarly, a 40-nm ETL TiO$_2$ layer for the extraction of electrons was coated on the substrate before the FACsPbI$_3$/ PMMA deposition for Sample C. Control experiments have been conducted to ensure that the PL is quenched in both Samples B and C due to the extraction of holes and electrons, respectively (Supplementary Fig. 8). Note that the charge dynamics in the transport layers (~300 nm below the surface) would not affect the iMIM results due to the shallow probing depth of 50–100 nm. The tr-iMIM data in Samples B and C under the 446-nm laser illumination with $P_C = 100 \, \text{mW/cm}^2$ are shown in Fig. 3a. It is evident that only the fast process with $\tau_1 \sim 0.7 \, \mu s$ survives in Sample B and the slow process with $\tau_2 \sim 10 \, \mu s$ in Sample C. The observation strongly suggests that the two time constants in Sample A are associated with the lifetime of electrons and holes in FACsPbI$_3$.

The HTL/ETL samples also allow us to separately address the diffusion dynamics of electrons and holes. Figure 3b, c show the tip-scan photoconductivity maps of Samples B and C under $P_C = 100 \, \text{mW/cm}^2$, from which $L_e \sim 5.2 \, \mu m$ and $L_h \sim 2.7 \, \mu m$ can be extracted, respectively. As tabulated in Fig. 3d, we can derive the diffusion coefficient from the diffusion equation $L = \sqrt{D\tau}$ and carrier mobility ($\mu_{e,diff} = 15 \, \text{cm}^2/\text{V·s}$ and $\mu_{h,diff} = 0.3 \, \text{cm}^2/\text{V·s}$) from

the Einstein relation $\mu = (q/k_BT)*D$. A different method to analyze the transport properties is through the photoconductivity (Eq. 1) and density profile (Eq. 2). The calculated mobility values are $\mu_{e,pc} = 24 \, \text{cm}^2/\text{V·s}$ and $\mu_{h,pc} = 0.3 \, \text{cm}^2/\text{V·s}$. The small difference between the two methods is within the error bars of the measurements. We note that in MAPbI$_3$ and FAPbI$_3$ thin films, mobility values measured by different techniques vary in a considerable range from 0.2 to 30 cm$^2$/V·s[6–10,14,17,23–25,28,40,41]. As tabulated in Supplementary Fig. 9, either $\mu_e > \mu_h$ or $\mu_e < \mu_h$ has been reported in the literature. In our experiment, mobility values are directly calculated from the measured $L$ and $\tau$ under an illumination intensity ~1 Sun, with no other assumptions or modeling involved. The pronounced difference between $\mu_e$ and $\mu_h$ is thus genuine. Figure 3d also indicates that the equilibrium carrier density in our experiment is on the order of $10^{15}$–$10^{16} \, \text{cm}^{-3}$. Within this range, the electron/ hole mobility is largely independent of the carrier concentration[42]. It is thus well justified to approximate the density profile by the measured photoconductivity profile in our diffusion analysis (Fig. 1e).

Finally, we briefly discuss the iMIM results at higher illumination intensities. As shown in Fig. 4a, b, the temporal evolution of HTL/ETL samples again displays the biexponential decay when $P_C$ increases beyond 100 mW/cm$^2$ (complete data in Supplementary Fig. 10), with one of the processes substantially suppressed. For instance, while $A_1/A_2 \sim 2$ is expected in plain FACsPbI$_3$, the electron dynamics clearly dominate in Sample B such that $A_1/A_2 > 2$ in Fig. 4c. Conversely, with electrons efficiently removed by the ETL, the hole dynamics prevail and $A_1/A_2 < 2$ in Sample C. We have also performed photoconductivity mapping on Samples B and C under various $P_C$ (Supplementary Figs. 11 and 12) and the results are plotted in Fig. 4d. As $P_C$ increases towards $10^3$–$10^4 \, \text{mW/cm}^2$, the contribution from the other type of carriers is no longer negligible. Consequently, in addition to the general trend of decreasing diffusion length at increasing excitation, $L$ in Sample B decreases further at high $P_C$, whereas $L$ in Sample C increases slightly at high $P_C$. Similarly, while only one type of carriers is responsible for the low-power photoconductivity, $\sigma_C$ does not scale with $P_C$ in either sample towards $10^4 \, \text{mW/cm}^2$ (Fig. 4e).

## Discussion

The spatiotemporally resolved iMIM experiments reveal rich information on organometal trihalide perovskite thin films. To begin with, we take a close look at the impact of GBs on charge transport in PSC materials, which has been under intense debate[19,43–46]. As summarized in a recent review[47], while GBs strongly affect the current–voltage hysteresis and long-term stability of PSCs, their effect on carrier recombination and thus the open-circuit voltage is rather mild under the illumination of ~1 Sun. In a previous report[33], we showed that the photoconductivity is spatially homogeneous over grains and GBs, consistent with conductive AFM and SPCM studies[20,46]. In this work, we further demonstrate that the carrier diffusion is not impeded by the presence of numerous GBs in all three samples. It is possible that the GBs in the current study are not strong nonradiative recombination (i.e., highly defective) centers, and there is no significant band bending at the GBs to block electron/ hole movement across multiple grains[48]. As a result, under the normal solar-cell operation, GBs in our samples do not lead to appreciable spatial variation of transport properties such as the density and mobility of photoexcited carriers, consistent with the early theoretical prediction[49]. We caution that sample-to-sample variation is widely observed in the PSC research. It is still possible that GBs in other hybrid perovskite thin films exhibit strong impacts on the carrier lifetime and transport properties.

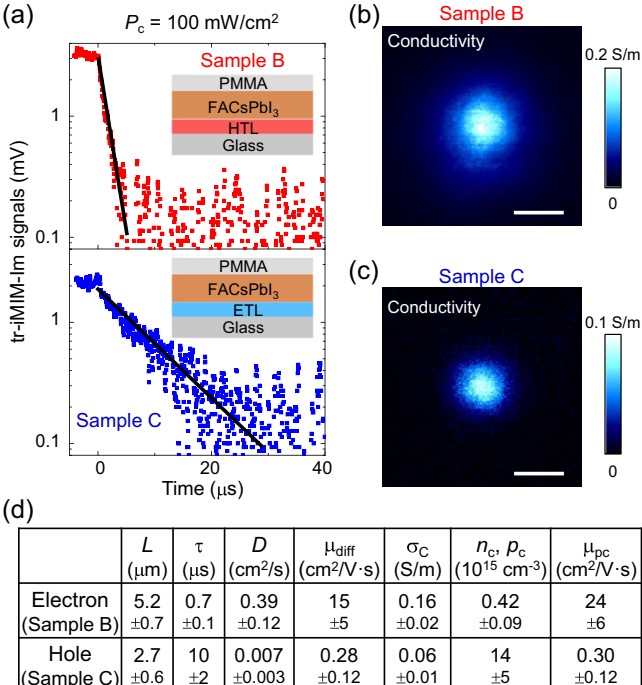

**Fig. 3 Results and analysis on HTL/ETL samples. a** tr-iMIM signals on the HTL Sample B (upper panel) and ETL Sample C (lower panel). The layer structures of each sample are illustrated in the insets. **b** Diffusion maps of Sample B and **c** Sample C under the illumination of 446-nm laser at $P_C = 100 \, \text{mW/cm}^2$. Scale bars are 10 μm. **d** Tabulated parameters for the calculation of electron/hole mobility values by two methods, i.e., $\mu_{diff}$ from the Einstein relation and $\mu_{pc}$ from photoconductivity analysis.

|  | $L$ (μm) | $\tau$ (μs) | $D$ (cm$^2$/s) | $\mu_{diff}$ (cm$^2$/V·s) | $\sigma_C$ (S/m) | $n_c$, $p_c$ ($10^{15}$ cm$^{-3}$) | $\mu_{pc}$ (cm$^2$/V·s) |
|---|---|---|---|---|---|---|---|
| Electron (Sample B) | 5.2 ±0.7 | 0.7 ±0.1 | 0.39 ±0.12 | 15 ±5 | 0.16 ±0.02 | 0.42 ±0.09 | 24 ±6 |
| Hole (Sample C) | 2.7 ±0.6 | 10 ±2 | 0.007 ±0.003 | 0.28 ±0.12 | 0.06 ±0.01 | 14 ±5 | 0.30 ±0.12 |

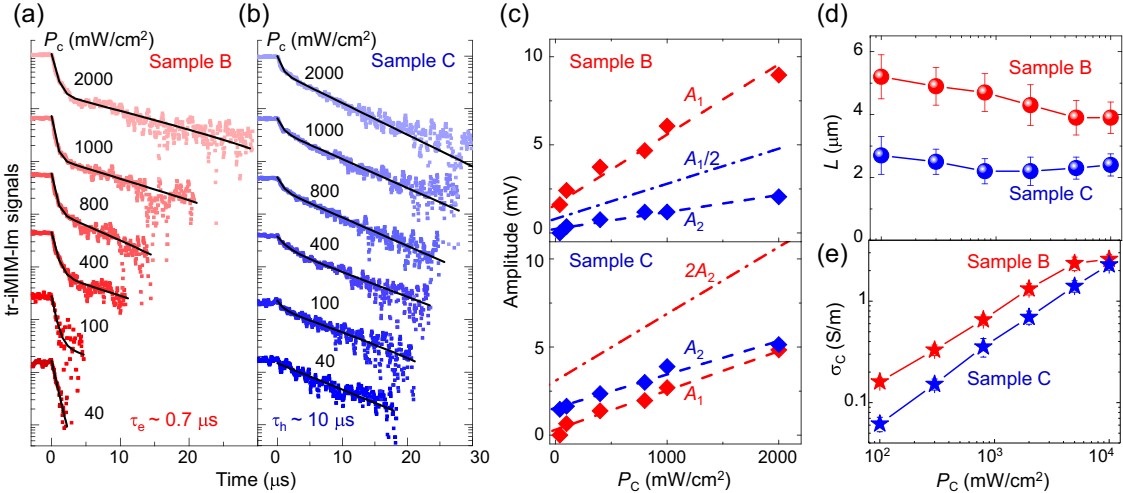

**Fig. 4 Power-dependent iMIM results. a** tr-iMIM signals on Sample B and **b** Sample C under various laser powers. Signals below the noise level are truncated for clarity. **c** Power-dependent $A_1$ and $A_2$ in Sample B (upper panel) and Sample C (lower panel). The dash-dotted lines are $A_1/2$ (Sample B) and $2A_2$ (Sample C) for comparison with the plain perovskite Sample A, which shows $A_1/A_2 \sim 2$. **d** Power-dependent diffusion lengths and **e** photoconductivity at the center of the illumination spot in both samples.

Given the extensive use of PL in studying carrier dynamics, it is instructive to compare the TRPL and tr-iMIM results in our samples. In short, TRPL measured excited states such as exciton recombination via emitted photons, whereas tr-iMIM measures the decay of steady-state conductivity following optical injection of free carriers. In TRPL experiments, the signal strength depends on the radiative recombination process that emits photons, whereas the temporal evolution measures the total lifetime of certain carriers or excitons[8–10,18,19]. For the TRPL data of Sample A in Fig. 2b, the fast ($\tau_f \sim 150$ ns) and slow ($\tau_s \sim 0.7$ μs) processes are associated with the surface recombination and the relaxation of the shorter-lived electrons in FACsPbI$_3$, respectively. Note that $\tau_s$ matches $\tau_1$ in the tr-iMIM data. After the elapse of $\tau_s$, however, no more mobile electrons are available for radiative recombination with mobile holes. As a result, TRPL cannot measure the lifetime of the longer-lived carriers[16]. We emphasize that the extraction of one type of carriers by HTL or ETL quenches the PL process and the TRPL decay constants in these samples no longer represent lifetimes of electrons or holes in plain PSCs[50]. In tr-iMIM, however, both the signal strength and temporal evolution depend on photoconductivity, which is proportional to the product of carrier density and mobility. For the three time scales above, the decay on the order of 100 ns is not seen in tr-iMIM, presumably due to the small steady-state density and low mobility of surface-bound carriers. On the other hand, because of the low efficiency of radiative recombination in PSCs[16], the removal of free electrons does not lead to appreciable changes in the dynamics of free holes. Consequently, the relaxation process of electrons and holes can be treated independently, as revealed by the tr-iMIM data. It should be noted that PL microscopy has also been utilized to map out the diffusion dynamics in PSC materials[51,52]. For the same reasons discussed above, it is not straightforward to compare photoluminescence and photoconductivity imaging results across multiple grains, which will be subjected to future experiments.

The difference between photo-physical properties of electrons and holes, as evidenced in Fig. 3d, deserves further discussions. In hybrid perovskites, deep-level defects dominate the trapping/de-trapping process and nonradiative recombination of free carriers. In general, the deeper the trap level, the longer time it takes for carriers to be de-trapped, and consequently the longer lifetime and lower mobility. Theoretical studies[53,54] suggest that cation

and anion vacancies create shallow energy levels, while iodine interstitials introduce deep levels in the bandgap. Interestingly, iodine interstitials can be both positively ($I_i^+$) and negatively ($I_i^-$) charged, which leads to spatially separated trapped electrons and holes with very low recombination efficiency. The transition energy for $I_i^+$ (0/+) (de-trapping for electron) is calculated to be 0.48 eV below the conduction band minimum (CBM), whereas the transition energy for $I_i^-$ (0/−) (de-trapping for hole) is 0.78 eV above the valence band maximum (VBM)[54]. The larger de-trapping barrier for hole results in its longer lifetime and lower mobility. When photoexcited electrons are quenched, the remaining holes will be trapped and then de-trapped via $I_i^-$, and vice versa. The trapping/de-trapping process induces delayed recombination, as manifested in the tr-iMIM decay curves. This qualitatively explains that the holes have a long carrier lifetime but lower mobility than electrons. Further theoretical work is needed to elucidate this physical picture in a quantitative manner.

As a final remark, we emphasize that in solar cells, diffusion lengths of both electrons and holes much larger than the film thickness is desirable for the effective separation of photocarriers. Because of the unique defect properties in hybrid perovskite thin films, as well as the competition between the recombination and trapping/de-trapping process, the imbalance in mobility ($\mu_e \gg \mu_h$) is largely compensated by the imbalance in lifetime ($\tau_e \ll \tau_h$). As a result, the difference between $L_e \sim 5$ μm and $L_h \sim 3$ μm is insignificant in our samples, which is of fundamental importance for the superior performance of PSC devices.

In summary, we systematically study the optoelectronic properties of 5%-Cs-doped FAPbI$_3$ thin films (PCE > 20%) by imaging the carrier diffusion in real space and detecting the photoconductivity evolution in real time. For plain perovskite films, two relaxation processes are observed on the sample. By selectively removing one type of carriers, we demonstrate that the fast and slow decay constants are associated with the lifetimes of photo-generated electrons and holes, respectively. The diffusion mapping on HTL/ETL samples allows us to extract parameters such as diffusion coefficient, equilibrium carrier density, and mobility of both carriers. The imbalance in carrier lifetime is offset by the difference in mobility such that diffusion lengths of electrons and holes are comparable to each other. We emphasize that, prior to our work, separate experiments are needed to

measure relaxation time (TRPL or TRTS) and mobility (transport or SPCM on doped samples) of free carriers. To our knowledge, it is the first time that diffusion length, carrier lifetime, and charge mobility can be individually addressed for mobile electrons and holes on the same batch (as-grown, HTL-coated, and ETL-coated) of samples. The spatiotemporal microwave imaging provides the most direct measurement of photo-physical properties of organometal trihalides, which is crucial for the research and development of these fascinating materials towards commercial products.

## Methods

**Materials**. All solvents were purchased from Sigma-Aldrich and used as-received without any other refinement. Formamidinium iodine (FAI) was purchased from Greatcell Solar. Lead iodide (PbI$_2$) was from TCI Corporation. Spiro-OMeTAD was received from Merck Corporation. Cesium iodine (CsI) and PTAA were purchased from Sigma-Aldrich. Patterned fluorine-doped tin-oxide-coated (FTO) glass (<15 Ω/square) and indium tin-oxide-coated (ITO) glass were obtained from Advanced Election Technology Co., Ltd.

**Sample preparation**. The perovskite films in this work were deposited on top of cover glasses or ITO glass. The substrate glasses were cleaned extensively by deionized water, acetone, and isopropanol. For the HTL deposition, the PTAA (Sigma-Aldrich) was dissolved in toluene with a concentration of 5 mg mL$^{-1}$ and spin-coated on the substrates at 5000 rpm for 30 s. The spun PTAA films were annealed at 100 °C for 10 min. For the ETL deposition, a compact titanium dioxide (TiO$_2$) layer of about 40 nm was deposited by spray pyrolysis of 7 mL 2-propanol solution containing 0.6 mL titanium diisopropoxide bis(acetylacetonate) solution (75% in 2-propanol, Sigma-Aldrich) and 0.4 mL acetylacetone at 450 °C in air. The FA$_{0.95}$Cs$_{0.05}$PbI$_3$ precursor solution was prepared by dissolving 0.4 M Pb$^{2+}$ in dimethyl sulfoxide (DMSO) and dimethylformamide (v/v = 3/7) mixed solvent. Perovskite films were deposited using a three-step spin-coating procedure with the first step of 100 rpm for 3 s, the second step of 3500 rpm for 10 s, and the last step of 5000 rpm for 30 s. Toluene (1 mL) was applied on the spinning substrates at 20 s of the third step. After spin coating, the substrates were annealed at 170 °C for 27 min. The encapsulated perovskite films were capped with PMMA (Mw about 120,000) film by spin-coating 15 mg ml$^{-1}$ PMMA in chlorobenzene solution at 4000 rpm for 35 s.

**iMIM and tr-iMIM setup**. The sample-scan iMIM was performed on a modified ParkAFM XE-100 platform with bottom illumination. The tip-scan iMIM was performed in a customized chamber (ST-500, Janis Research Co.) with positioners and scanners (AttoCube Systems AG). During the measurements, we kept the samples in a vacuum by pumping the chamber below 10$^{-4}$ mbar. The PtIr tips were purchased from Rocky Mountain Nanotechnology LLC, model 12PtIr400A for diffusion mapping and 12PtIr400A-10 (ultra-sharp tips) for the point measurements in Fig. 2. For tr-iMIM, the diode laser was modulated by an EOM (M350-160–01 EOM, Conoptics Inc.) with a power supply of 8-ns rise/fall time. The EOM was driven by a 7-kHz square wave from a function generator (DG5071, RIGOL Technologies USA Inc.) with <4-ns rise/fall time. The tr-iMIM signals were measured by a 600-MHz oscilloscope (DS6062, RIGOL Technologies USA Inc.) with a 5-GSa/s sampling rate.

**Finite-element analysis**. FEA was performed using the AC/DC module of commercial software COMSOL4.4. The tip for diffusion mapping is relatively blunt due to the extensive contact-mode scans on the sample surface. We modeled it as a truncated cone with a half-angle of 15° and a diameter of 200 nm at the apex. The ultra-sharp tip was mostly used for point measurements in Fig. 2 and the apex was well preserved. It was thus modeled as a truncated cone with a half-angle of 38° and a diameter of 20 nm at the apex. In the simulation, the dielectric constants of PMMA and FACsPbI$_3$ are 3 and 62, respectively, consistent with that reported in the literature[33,55]. The FEA software computes the real and imaginary parts of the tip-sample admittance (proportional to iMIM-Re/Im signals) as a function of the conductivity of the perovskite layer, using the values at σ = 0 as the reference.

## Data availability

All data supporting the findings of this study are available within the article and/or the SI Appendix. The raw data is available from the corresponding author upon reasonable request.

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

## Acknowledgements

The research at UT-Austin was primarily supported by the NSF through the Center for Dynamics and Control of Materials, an NSF Materials Research Science and Engineering Center (MRSEC) under Cooperative Agreement DMR-1720595. The authors also acknowledge the use of facilities and instrumentation supported by the NSF MRSEC. K.L. and X.M. acknowledge the support from Welch Foundation Grant F-1814. X. Li acknowledges the support from Welch Foundation Grant F-1662. The tip-scan iMIM setup was supported by the US Army Research Laboratory and the US Army Research Office under Grants W911NF-16-1-0276 and W911NF-17-1-0190. The work at NREL was supported by the US DOE under Contract No. DE-AC36-08GO28308 with Alliance for Sustainable Energy, Limited Liability Company (LLC), the Manager and Operator of the National Renewable Energy Laboratory. K.Z., J.H., X.C., X.W., and Y.Y. acknowledge the support on charge carrier dynamics study from the Center for Hybrid Organic-Inorganic Semiconductors for Energy (CHOISE), an Energy Frontier Research Center funded by the Office of Basic Energy Sciences, Office of Science within the US DOE. F.Z. acknowledges the support on devices fabrication and characterizations from the De-Risking Halide PSCs program of the National Center for Photovoltaics, funded by the US DOE, Office of Energy Efficiency and Renewable Energy, Solar Energy Technologies Office. The views expressed in the article do not necessarily represent the views of the DOE or the US Government. The US Government retains and the publisher, by accepting the article for publication, acknowledges that the US Government retains a nonexclusive, paid-up, irrevocable, worldwide license to publish or reproduce the published form of this work or allow others to do so, for US Government purposes.

## Author contributions

K.Z. and K.L. conceived the project. F.Z. and J.H. prepared samples and characterized devices. X.C. and Z.H. performed the TRPL and PL characterization. X.M. and Z.C. performed the iMIM experiments and data analysis. J.Q. and X.L. contributed to the sample-scan experiments. X.W. and Y.Y. contributed to the theoretical explanations. X.M. and K.L. drafted the manuscript with contributions from all authors. All authors have given approval to the final version of the manuscript.

## Competing interests

The authors declare no competing interests.
