## [Peer Review File · Nature Communications]

REVIEWER COMMENTS

Reviewer #1 (Remarks to the Author):

In this paper entitled 'Superior Photo-carrier Diffusion Dynamics in Organic-inorganic Hybrid Perovskites Revealed by Spatiotemporal Conductivity Imaging', the authors utilized an emerging pump-probe microwave impedance microscopy to investigate the spatially resolved charge carrier diffusion dynamics in lead halide perovskite films. Although the technique is impressive, a similar methodology has already been used in the author's previous paper in ref. 33 where the conclusion on the in-homogenous distribution of the photoconductivity over the film structure is also the same. Therefore the novelty and significance of this paper are highly undermined.

In this paper, the authors claimed that the main new finding is the observation of inhomogeneous charge carrier diffusion dynamics over the perovskite film regardless of the grain or GB area. However, the fatal issue is the iMIM method used in this work has only 10 ns time resolution while lots of the major processes, such as trapping by the defects state in the grain or at the GB are most likely happen within 10 ns. In addition, given the standard mobility or diffusion coefficient of lead halide perovskites, the photo-generated carriers can easily diffuse for hundreds nm or several um within 10 ns time, (see the evidence from PL microscopy measurement in literature, for example, J. Am. Chem. Soc., 2015, 137, 12458–12461; Nature Mater. | 2020, 19 , 412–418. As the sample's grain size in this paper is around several hundreds of nm as shown in Fig. 2C, I would expect the charge carrier to diffuse around already and reach an equilibrium spatial distribution across the grain and GB after 10 ns. Therefore the dynamics mapping at different spot of the sample cannot represent the full photophysical processes of the charge carrier after excitation.

Another main problem is the signal of the iMIM measurement is the transient photoconductivity which is the combination between the mobility and concentration of the carriers. It is unclear that the kinetics of the signal showing in the paper represent only the decay of the carrier population or also combined with decay of carrier mobility. This is vital as the charge mobility can also be influenced by the scattering to the defects at the GB, which maybe happen at long time-scale. In order to extract the intrinsic carrier diffusion dynamics, the variation of the mobility vs. time should first be deconvoluted.

Based on the above concerns, I reject this manuscript and also offer some minor comments for the authors.

- 1) I am not so convinced by the claim that the charge carrier generation is homogenous along the vertical direction as the thickness of the film is larger than the absorption depth according to the authors in the paper. This means an initial inhomogeneous distribution of the charge carrier vertically cannot be avoided. Extra explanations on this point may be necessary.
- 2) How large is the laser beam size? This parameter is essential for the analysis of the spatially resolved conductivity.
- 3) The authors reported a drastically different mobility between electrons and holes (2.4 versus 0.3 cm²/Vs) by about one order of magnitude, which is not so commonly observed in other reports. Can such a difference be explained since the intrinsic effective mass of electrons and holes in lead halide perovskites are very similar?
- 4) In Fig. 2a, why the signal at the early time-scale after excitation is flat? Is it related to the response function of the device or something meaningful for the carrier dynamics?
- 5) I suggest to measure the PL of sample B and sample C to check if the emission has been totally quenched by the HTL/ETL, which is a standard protocol to verify the complete charge transfer.

Reviewer #2 (Remarks to the Author):

This manuscript reports the mobilities and lifetimes of hole and electron in 5% Cs-doped FAPbI₃ perovskite by using time-resolved light-illuminated microwave conductivity microscope (iMIM). By quenching hole or electron in underneath electron transport layer (ETL) or hole transport layer (HTL), the authors successfully separate the contribution of hole and electron and found the contrast in their mobility and lifetime. Of interest, holes show a low mobility (0.3 cm²/Vs) and long lifetime (10 us), while electrons show a high mobility (10 cm²/Vs) and short lifetime (0.7 us), which leads to the relatively balanced diffusion length (3-5 um). Given the unique measurements and convincing analysis (diffusional theory, power dependence, statistics, etc) as well as the insightful results on the nature of charge carrier dynamics in perovskite solar cells, I believe that this manuscript is suitable for publication in Nature Communication.

Minor

- 1) Superscript “-1” of 5 mg ml⁻¹ in Sample preparation
- 2) Remove hyphen (-) and put a space in 7-mL, 0.6-mL, and 0.4-mL in Sample preparation

Reviewer #3 (Remarks to the Author):

The manuscript describes the free-carrier dynamics in grain and boundaries of Cs-doped FAPbI₃ thin films by microwave imaging with 10 ns and 20 nm resolutions. The two relaxation times on the order of 1 μs and 10 μs are assigned to be the lifetimes of electrons and holes in FACsPbI₃, respectively. There is no substantial contrast for grain and grain boundaries. The technique is a powerful one to investigate carrier diffusion in photovoltaic materials and measurements are interesting. However there are a few concerns and confusions that have to be addressed before considering publication.

My main concern: Although the scanning tip is sharp the optical excitation is still diffraction limited. It is not clear how this hybrid configuration will affect the spatial and time resolution. Some more detailed analysis and controlled measurements will be useful to determine the simultaneous time and spatial resolution. For example, the lifetimes measured maybe affected by the density ingredient created by the micrometer excitation spot size. Diffusion length measured may also be affected by the micrometer excitation spot size since carriers within the excitation spot travel with different length.

If the technique indeed has the simultaneous 20 nm spatial resolution and ns time resolution, that should provide evidence for the difference in the carrier transport between dark and bright grain and boundaries. It has been measured at various investigations that there exists bright and dark grains in PL and grain boundaries with different lifetimes and transports. If they are indeed no difference, it will be interesting to discuss the origin in more details and reconcile different reports.

In general, I'm still confused why grain boundary and grain have the same decay times? They should be different as measured before using TRPL. Grain boundary and grain should also have very different diffusion coefficients which produce enough conductivity contrast.

How to protect the sample from degradation? It looks like the samples are held in some chambers. It will be benefit for the readers for more information on the setup, especially more details on sample/tip/chamber configuration.

What determines the decay times in tr-iMIM: intrinsic bandgap e/h recombination or charge separation/moving from the excitation spots? It will be beneficial to clarify exactly what is measured.

Much longer diffusion length and lifetimes have been reported before in various reports. How to reconcile them with the values reported here, e.g., diffusion length ~3-5 um?

What is the time scales for the trapping and de-trapping processes discussed? Can tr-iMIM resolve those times?

Figure 2b: it is not clear for me if the data at each pixel are generated by time-integrated signals or at fixed time interval between optical excitation and microwave pulse. I'm confused since it will take time for carriers to diffuse outwards to reach the outside. That doesn't seem to be considered in the simple diffusion model, eqs. 1-3.

Some more minor questions that may deserve some further clarification:

Is the excitation used close to 1 sun?

How do authors differentiate conductivity contrast vs topography contrast?

Fig. S5: I expect different excitation energy will produce different diffusion length since hot carriers with different temperature will travel differently especially across grains.

In summary, the manuscript contains interesting data using a powerful spatial-temporal imaging technique. It will be much more informative if the above questions can be addressed.

Reply to Reviewer 1's report:

The Reviewer raised several technical questions on the iMIM measurements. Before addressing them in full, it is instructive to review the underlying physics of our method. Our **steady-state** iMIM experiment (Fig. 1) addresses the most relevant photophysical process in solar cells, i.e., the transport of photo-generated mobile carriers under the continuous illumination of ~ 1 Sun. Here the equilibrium carrier density is directly proportional to their average lifetime in the conduction/valence band. Free-carrier lifetimes in many semiconductors, including the hybrid perovskites, are determined by relatively slow (ns to ms) processes such as carrier detrapping from defects or phonon-assisted nonradiative recombination, which are the relevant time scales in our **time-resolved** photoconductivity studies (Figs. 2 and 3). On the other hand, photoluminescence (PL) measurement, as mentioned multiple times by the Reviewer, is an optical technique that detects photons emitted from the radiative recombination of bound electrons and holes. It is widely used to address the fast dynamics, e.g., carrier generation, radiative recombination, and carrier trapping, in photovoltaics. In short, PL measured excited states such as exciton recombination via emitted photons, whereas iMIM measures near-equilibrium conductivity following optical injection of free carriers. We believe that, by articulating the difference between photoluminescence and photoconductivity experiments, we will be able to avoid confusions when addressing comments from Reviewer 1.

Reviewer Comment #1-1: *In this paper entitled ‘Superior Photo-carrier Diffusion Dynamics in Organic-inorganic Hybrid Perovskites Revealed by Spatiotemporal Conductivity Imaging’, the authors utilized an emerging pump-probe microwave impedance microscopy to investigate the spatially resolved charge carrier diffusion dynamics in lead halide perovskite films. Although the technique is impressive, a similar methodology has already been used in the author’s previous paper in ref. 33 where the conclusion on the in-homogenous distribution of the photoconductivity over the film structure is also the same. Therefore the novelty and significance of this paper are highly undermined.*

Reply #1-1: We are delighted that the Reviewer is impressed by our technique. The novelty and broad impact of our work compared with previously published results are justified as follows. For **steady-state** measurements, our earlier sample-scan work (Ref. 33) only probed the photoconductivity on the excitation point. This work, on the other hand, makes a major step forward to map out the diffusion dynamics away from the laser spot. For **time-resolved** measurements, we separately identified the temporal dynamics of free electrons and holes, which provides crucial information for solar-cell operations. As a result, while we also come to the same conclusion on local photoresponse as that in Ref. 33, the focus of this work is the first noncontact spatiotemporal photoconductivity imaging of both electrons and holes in hybrid perovskites. The novelty and significance of this work is thus *NOT* undermined by our prior publication.

Review Comment #1-2: *In this paper, the authors claimed that the main new finding is the observation of inhomogeneous charge carrier diffusion dynamics over the perovskite film*

regardless of the grain or GB area. However, the fatal issue is the iMIM method used in this work has only 10 ns time resolution while lots of the major processes, such as trapping by the defects state in the grain or at the GB are most likely happen within 10 ns. In addition, given the standard mobility or diffusion coefficient of lead halide perovskites, the photo-generated carriers can easily diffuse for hundreds nm or several μm within 10 ns time, (see the evidence from PL microscopy measurement in literature, for example, *J. Am. Chem. Soc.*, 2015, 137, 12458–12461; *Nature Mater.* 2020, 19, 412–418. As the sample's grain size in this paper is around several hundreds of nm as shown in Fig. 2C, I would expect the charge carrier to diffuse around already and reach an equilibrium spatial distribution across the grain and GB after 10 ns. Therefore the dynamics mapping at different spot of the sample cannot represent the full photophysical processes of the charge carrier after excitation.

Reply #1-2: As stated in our opening remarks, the iMIM diffusion mapping is a steady-state measurement of photoconductivity under the dynamic equilibrium. We did not attempt to track the motion of carriers within the 10 ns time resolution, as questioned by the Reviewer. Instead, we were mapping the equilibrium diffusion patterns of photo-generated electrons and holes under CW laser and fitting the photoconductivity profile to the diffusion equation, which is a standard process to extract the diffusion length (see Ref. 27, 37, 38).

For the time-resolved iMIM measurements, we completely agree with the Reviewer that the time scale of < 10 ns such as trapping by defect states in the grain or at the GB is beyond our temporal resolution. However, the lifetime of mobile carriers in this material is determined by detrapping of electrons (holes) back to the conduction (valence) band, which has a rather long (μs) time scale. This is widely studied by other microwave conductivity experiments (Refs. 22 – 25) in a non-spatially-resolved manner. We would also like to point out that the tr-iMIM measurement is performed by using square-wave-driven CW laser (see Fig. 1a) such that the photoconductivity reaches the steady state at $t = 0$. We will elaborate on this aspect in Reply #1-7.

Finally, we thank the Reviewer for bringing up the references on PL microscopy. We agree that charge carriers have already diffused around grains (hundreds of nm) and GBs after the 10ns time scale, which is completely consistent with our steady-state diffusion data. We caution that it is not straightforward to compare PL microscopy and iMIM results. PL harvests the photon emitted during the radiative decay of excitons, whereas the iMIM measures the free-carrier transport. Such a comparison requires much more control experiments and is beyond the scope of our work.

Changes #1-2: We include the two references suggested by the Reviewer as Refs. 50 and 51 in the revised manuscript. Accordingly, we add two sentences on Page 9 as “It should be noted that PL microscopy has also been utilized to map out the diffusion dynamics in PSC materials^{50,51}. For the same reasons discussed above, it is not straightforward to compare photoluminescence and photoconductivity imaging results across multiple grains, which will be subjected to future experiments.”

Review Comment #1-3: *Another main problem is the signal of the iMIM measurement is the transient photoconductivity which is the combination between the mobility and concentration of the carriers. It is unclear that the kinetics of the signal showing in the paper represent only the decay of the carrier population or also combined with decay of carrier mobility. This is vital as*

the charge mobility can also be influenced by the scattering to the defects at the GB, which maybe happen at long time-scale. In order to extract the intrinsic carrier diffusion dynamics, the variation of the mobility vs. time should first be deconvoluted.

Reply #1-3: The Reviewer raised a valid point in this comment. The photoconductivity determined from iMIM signals is proportional to both the carrier density (n) and mobility (μ). In the general case, one would need to know the $n - \mu$ relation in order to deconvolve the density profile from the iMIM data. On the other hand, it is widely accepted that for typical hybrid perovskite thin films, electron/hole mobility is largely independent of the carrier concentration for n or p below 10^{18} cm^{-3} (see, e.g., Motta et al. “Charge carrier mobility in hybrid halide perovskites” *Sci. Rep.* **5**, 12746 (2015)). As tabulated in Fig. 3d, the equilibrium carrier concentration under ~ 1 Sun illumination is on the order of 10^{15} – 10^{16} cm^{-3} in our experiment. As a result, the intrinsic electron/hole mobility does not depend on the charge density in this regime. We agree that the local mobility values could be affected by scattering at the GBs. However, we did not observe sudden changes of signals across the GBs from our iMIM diffusion data. In fact, if the mobility is a strong function of carrier densities (e.g., strong power-law or exponential relationship) or changes across GBs, we would not have the excellent fitting (statistical R^2 -value > 0.98) to Eq. (3) in Fig. 1e. In summary, we believe that our statement on Page 5 “Assuming that the carrier mobility μ is independent of charge density n within the range of our experiment” is a good approximation for the analysis of iMIM diffusion data.

Changes #1-3: We cite the paper mentioned above as Ref. 49 to justify the independence of carrier mobility on charge density within the range of our experiment. Accordingly, we add the following statements on Page 7 of the revised manuscript – “Fig. 3d also indicates that the equilibrium carrier density in our experiment is on the order of $10^{15} - 10^{16} \text{ cm}^{-3}$. Within this range, the electron/hole mobility is largely independent of the carrier concentration⁴⁹. It is thus well justified to approximate the density profile by the measured photoconductivity profile in our diffusion analysis (Fig. 1e).”

Review Comment #1-4: *1) I am not so convinced by the claim that the charge carrier generation is homogenous along the vertical direction as the thickness of the film is larger than the absorption depth according to the authors in the paper. This means an initial inhomogeneous distribution of the charge carrier vertically cannot be avoided. Extra explanations on this point may be necessary.*

Reply #1-4: We agree with the Reviewer that, strictly speaking, in the time scale relevant to electron-hole generation ($\text{fs} \sim \text{ps}$), the charge density in the vertical direction is not uniform across the film thickness. However, as explained in our opening remarks, the steady-state iMIM measurement in Fig. 1 is taken under CW laser illumination and the tr-iMIM measurement in Fig. 2 has a temporal resolution of 10 ns. In both cases, the photo-generated carriers would have diffused vertically across the 250 nm-thick film. Consequently, we believe that, for all practical purposes in this experiment, vertical inhomogeneity of charge carriers is negligible.

Changes #1-4: In order to clarify this point, we modify the sentence on Page 4 to be “...such that the photoconductivity is uniformly distributed in the vertical direction within the relevant time scale in our experiment.”

Review Comment #1-5: 2) *How large is the laser beam size? This parameter is essential for the analysis of the spatially resolved conductivity.*

Reply #1-5: The size of the laser beam was quantitatively described on Page 5 of our manuscript as "... the Gaussian beam profile (e^{-r^2/w^2} , $w \sim 2 \mu\text{m}$)". We agree that this parameter is essential for the analysis of spatially resolved photoconductivity.

Review Comment #1-6: 3) *The authors reported a drastically different mobility between electrons and holes (2.4 versus 0.3 cm²/Vs) by about one order of magnitude, which is not so commonly observed in other reports. Can such a difference be explained since the intrinsic effective mass of electrons and holes in lead halide perovskites are very similar?*

Reply #1-6: As stated in our manuscript and pointed out by the Reviewer, the intrinsic effective masses of electrons and holes in lead halide perovskites are very similar. The measured electron and hole mobilities in the literature, however, are drastically different from each other. In Fig. S9 of Supplementary Information, we tabulate a list of electron/hole mobility values from Refs. 6-10, 14, 17, 23-25, 28, 47, 48, where numbers ranging from $\mu_e \gg \mu_h$, $\mu_e \sim \mu_h$, to $\mu_e \ll \mu_h$ have been reported. It is quite obvious that the experimentally measured mobility is dependent on sample-specific extrinsic effects. Nevertheless, we emphasize that despite the disparity in their mobility, electrons and holes exhibit comparable diffusion lengths, which is likely due to the trapping/de-trapping process in our samples. In fact, on Page 9 of the manuscript, we have one full paragraph to discuss the possible deep-level traps in our hybrid perovskites, which qualitatively explains our experimental findings – "The difference between photo-physical properties of electrons and holes... qualitatively explains that the holes have longer carrier lifetime but lower mobility than electrons." We believe that the discussion of mobility disparity between electrons and holes is adequate in our manuscript.

Review Comment #1-7: 4) *In Fig. 2a, why the signal at the early time-scale after excitation is flat? Is it related to the response function of the device or something meaningful for the carrier dynamics?*

Reply #1-7: As stated on Page 5 of the manuscript and illustrated in Fig. 1a, we are using 7 kHz square waves to drive the electro-optic modulator (EOM). This is different from typical TRPL experiments using pulsed laser excitation. Both laser-ON and laser-OFF states are sufficiently long ($\sim 70 \mu\text{s}$ each) to reach steady-state and zero photoconductivity, respectively. As a result, the tr-iMIM signals before $t = 0$ in Figs. 2 and 3 are flat.

Changes #1-7: In order to avoid confusions, we revise the relevant sentence on Page 5 as "...such that steady-state photoconductivity is reached in the laser-ON state and zero photoconductivity in the laser-OFF state."

Review Comment #1-8: 5) *I suggest to measure the PL of sample B and sample C to check if the emission has been totally quenched by the HTL/ETL, which is a standard protocol to verify the complete charge transfer.*

Reply #1-8: We totally agree with the Reviewer that control experiments on Samples B and C are necessary to check the efficacy of the HTL and ETL, respectively. To this end, we performed PL measurements on Samples B and C and the results are included in the new Supplementary

Fig. S8. It is clearly observed that the PL signals are substantially quenched in samples with carrier transport layers. It should be noted that, in order to obtain good signal-to-noise ratio, the excitation power was kept high (on the order of ~ 100 Sun) in the PL measurements. As a result, the extraction of mobile carriers by HTL/ETL is not 100% complete, consistent with the high-power tr-iMIM data in Fig. 4. In addition, we would like to add Dr. Zhiyuan Huang, who conducted the PL experiments and participated in the manuscript revision process, into the author list.

Changes #1-8: We include the new PL data on all three samples in Fig. S8 of the Supplementary Information and a paragraph to explain the results. Accordingly, on Page 6 of the main text, we add one sentence and refer the readers to Fig. S8 for more details – “Control experiments have been conducted to ensure that the PL is quenched in both Samples B and C due to the extraction of holes and electrons, respectively (Supplementary Fig. S8).” We also include Dr. Zhiyuan Huang in the new author list.

Reply to Reviewer 2’s report:

Review Comment #2-1: *This manuscript reports the mobilities and lifetimes of hole and electron in 5% Cs-doped FAPbI₃ perovskite by using time-resolved light-illuminated microwave conductivity microscope (iMIM). By quenching hole or electron in underneath electron transport layer (ETL) or hole transport layer (HTL), the authors successfully separate the contribution of hole and electron and found the contrast in their mobility and lifetime. Of interest, holes show a low mobility (0.3 cm²/Vs) and long lifetime (10 us), while electrons show a high mobility (10 cm²/Vs) and short lifetime (0.7 us), which leads to the relatively balanced diffusion length (3-5 um). Given the unique measurements and convincing analysis (diffusional theory, power dependence, statistics, etc) as well as the insightful results on the nature of charge carrier dynamics in perovskite solar cells, I believe that this manuscript is suitable for publication in Nature Communication.*

Minor

1) Superscript “-1” of 5 mg ml⁻¹ in Sample preparation

2) Remove hyphen (-) and put a space in 7-mL, 0.6-mL, and 0.4-mL in Sample preparation

Reply #2-1: We are very grateful to the Reviewer’s compliments on our work and his/her explicit recommendation for publication after minor revisions. Changes are summarized below.

Changes #2-1: The formatting problems in the Sample preparation section have been corrected in the revised manuscript (5 mg·mL⁻¹ ... 7 mL ... 0.6 mL ...0.4 mL) following the Reviewer’s comments.

Reply to Reviewer 3’s report:

Review Comment #3-1:

The manuscript describes the free-carrier dynamics in grain and boundaries of Cs-doped FAPbI₃ thin films by microwave imaging with 10 ns and 20 nm resolutions. The two relaxation times on the order of 1 μs and 10 μs are assigned to be the lifetimes of electrons and holes in FACsPbI₃, respectively. There is no substantial contrast for grain and grain boundaries. The technique is a powerful one to investigate carrier diffusion in photovoltaic materials and measurements are interesting. However, there are a few concerns and confusions that have to be addressed before considering publication.

Reply #3-1: We thank the Reviewer for the nice summary of our work and his/her compliments on our new technique and the interesting measurements. We will address the concerns and confusions point-by-point as below.

Review Comment #3-2: *My main concern: Although the scanning tip is sharp the optical excitation is still diffraction limited. It is not clear how this hybrid configuration will affect the spatial and time resolution. Some more detailed analysis and controlled measurements will be useful to determine the simultaneous time and spatial resolution. For example, the lifetimes measured maybe affected by the density ingredient created by the micrometer excitation spot size. Diffusion length measured may also be affected by the micrometer excitation spot size since carriers within the excitation spot travel with different length.*

Reply #3-2: The Reviewer raised an important question on how the spatial resolution is defined in our iMIM technique. First of all, we completely agree with him/her that the optical excitation in our diffusion mapping is still diffraction limited. On the other hand, the profile of the illumination spot can be directly measured and fitted to a Gaussian shape, as shown in Fig. 1e. In other words, the lateral spread of the laser excitation is well-defined and can be quantitatively taken into account in the diffusion analysis. Secondly, we emphasize that we are measuring the **steady-state** photoconductivity profile in Fig. 1. In this case, the generation, recombination, and diffusion of carriers with finite lifetimes have reached the dynamic equilibrium condition, which is described by Eq. (2) in the main text and the solution given by Eq. (3). It is also evident that the micrometer-sized excitation spot is explicitly taken into account in Eqs. (2) and (3), which is a standard process to extract the diffusion length from measurements with diffraction-limited optical excitation (see Ref. 27, 37, 38). Thirdly, while there is an excitation gradient created by the micrometer excitation spot size, the laser power is kept low in Fig. 1 such that intensity-dependent processes (e.g., Auger) are unimportant. In fact, we demonstrated that the lifetime is largely power-independent up to ~ 20 Sun in Fig. S10. Finally, we agree with the Reviewer that we should specify the spatial and time resolution – in our case, the quoted 20-nm spatial resolution specifically refers to the *electrical* imaging in our system.

Changes #3-2: On Page 4 of the revised manuscript, we specify that “...with 20-nm spatial resolution and 10-ns temporal resolution for the electrical detection.” On Page 5, we also add a sentence to emphasize this point – “The optical excitation in our setup is diffraction limited, whereas the electrical imaging has a spatial resolution 20 ~ 50 nm comparable to the tip diameter.” We believe that this clarification is crucial to remove the ambiguity in our earlier statement.

Review Comment #3-3: *If the technique indeed has the simultaneous 20 nm spatial resolution and ns time resolution, that should provide evidence for the difference in the carrier transport between dark and bright grain and boundaries. It has been measured at various investigations that there exist bright and dark grains in PL and grain boundaries with different lifetimes and transports. If they are indeed no difference, it will be interesting to discuss the origin in more details and reconcile different reports. In general, I'm still confused why grain boundary and grain have the same decay times? They should be different as measured before using TRPL. Grain boundary and grain should also have very different diffusion coefficients which produce enough conductivity contrast.*

Reply #3-3: We thank the Reviewer for bringing up this question. Indeed, some investigations suggested that grains and grain boundaries exhibit very different lifetimes, while others indicated that the effect of GBs on transport properties is rather mild (see the recent review Ref. 44). In our steady-state diffusion mapping (Fig. 1), the spread of the photoconductivity signals ($L \sim 5 \mu\text{m}$) is clearly wider than the laser spot ($w \sim 2 \mu\text{m}$) and typical grain sizes ($< 1 \mu\text{m}$), indicative of free-carrier diffusion over many GBs. It should be noted that, since PL measures photons emitted from the radiative recombination, the signals will be quenched when either electrons or holes are trapped by GBs or other defects. In contrast, diffusion of photoconductivity over multiple grains can be detected as long as one type of free carriers passes through GBs without being trapped. Again, PL measured excited states such as exciton recombination via emitted photons, whereas iMIM measures near-equilibrium conductivity following optical injection of free carriers. The comparison between TRPL and tr-iMIM measurements has already been thoroughly discussed on Pages 8 – 9 of our manuscript.

With that said, we cannot rule out the possibility that the effect of GBs on lifetimes and transports is strongly sample-dependent. For instance, from our literature survey of the electron/hole mobility (see Fig. S9), it is clear that the photophysical properties in hybrid perovskite thin films vary significantly from samples to samples. In order to fully address this question, one needs to perform photoluminescence and photoconductivity microscopy on the same sample, ideally with the size of grains greater than that of the laser spot. Since this is far beyond the scope of this work, we would defer such experiments to future investigations.

Changes #3-3: On Page 8 of the revised manuscript, we removed a strong claim that “GBs in ~~hybrid organic inorganic perovskite thin films~~ do not lead to appreciable spatial variation of transport properties...” and changed it to “GBs in **our samples** do not lead to appreciable spatial variation of transport properties...”. We also ended the paragraph with the acknowledgement on sample-dependent properties – “**We caution that sample-to-sample variation is widely observed in the PSC research. It is still possible that GBs in other hybrid perovskite thin films exhibit strong impacts on the carrier lifetime and transport properties.**” In addition, on Page 9 of the revised manuscript, we include a statement on the difference between PL and tr-iMIM measurement – “**For the same reasons discussed above, it is not straightforward to compare photoluminescence and photoconductivity imaging results across multiple grains, which will be subjected to future experiments.**”

Review Comment #3-4: *How to protect the sample from degradation? It looks like the samples are held in some chambers. It will be benefit for the readers for more information on the setup, especially more details on sample/tip/chamber configuration.*

Reply #3-4: In both the main text and the “Sample preparation” section of Methods, we explained that the hybrid perovskite samples were protected by the capping layer of 30 nm PMMA. In the “iMIM and tr-iMIM setup” section of Methods, we further mentioned that “The tip-scan iMIM was performed in a customized chamber (ST-500, Janis Research Co.) ...”. Besides, the experiment was conducted in vacuum by pumping the ST-500 chamber, which is worth mentioning in the manuscript.

Changes #3-4: On Page 11 of the revised manuscript, we add one sentence to the Methods – “During the measurements, we kept the samples in vacuum by pumping the chamber below 10^{-4} mbar.”

Review Comment #3-5: *What determines the decay times in tr-iMIM: intrinsic bandgap e/h recombination or charge separation/moving from the excitation spots? It will be beneficial to clarify exactly what is measured.*

Reply #3-5: On Page 6 of the manuscript, we explained that, within the measurement range, the iMIM-Im signals are proportional to photoconductivity, which scales with both carrier density and mobility. Since the mobility is largely density-independent (see our Reply #1-3), the iMIM-Im signals are directly proportional to the local density of mobile carriers in the conduction or valence band. By fitting the relaxation of tr-iMIM signals to exponential decay curves, we obtain the lifetime of free electrons and/or holes, rather than the intrinsic bandgap recombination time or the time scale for charge separation/moving from the excitation spots. On Page 9 of the manuscript, we suggest that “...deep-level defects dominate the trapping/de-trapping process and nonradiative recombination of free carriers...”, which may explain the observed $1 \sim 10 \mu\text{s}$ time scale in our tr-iMIM measurement.

Changes #3-5: We agree with the Reviewer that it will be beneficial to clarify exactly what is measured in the tr-iMIM experiment. On Page 6 of the revised manuscript, we add a statement – “From Eq. (1), the decay of tr-iMIM-Im signal provides a direct measure of the lifetime of mobile carriers in the conduction or valence band.” – in order to elucidate our measurement.

Review Comment #3-6: *Much longer diffusion length and lifetimes have been reported before in various reports. How to reconcile them with the values reported here, e.g., diffusion length $\sim 3\text{-}5 \mu\text{m}$?*

Reply #3-6: We assume that the Reviewer was referring to the results reported in single-crystalline hybrid perovskite samples, e.g., Refs. 8, 16, where diffusion lengths up to $> 100 \mu\text{m}$ were reported. For solar-cell applications, solution-based thin-film samples with low fabrication cost are much more attractive. To our knowledge, on thin-film samples, the measured diffusion lengths for electrons and holes (e.g., Refs. 9, 10, 16, 46) are mostly on the order of $1 \sim 10 \mu\text{m}$. We believe that the diffusion lengths of $3 \sim 5 \mu\text{m}$ in our Cs-doped FAPbI₃ thin films are consistent with those reported in the literature.

Changes #3-6: On Page 6 of the revised manuscript, we add the citation to the literature as “...a diffusion length $L = 5.1 \pm 0.6 \mu\text{m}$, consistent with values reported in the literature for thin-film PSCs^{9,10,16,46}” to address the Reviewer’s comment.

Review Comment #3-7: *What is the time scales for the trapping and de-trapping processes discussed? Can tr-iMIM resolve those times?*

Reply #3-7: The generation and recombination of photo-carriers span a wide range of time scales. It takes many sophisticated tools (photoemission, optical pump-probe, etc.) to fully resolve these processes. In our case, the lifetime of mobile carriers in semiconducting PSCs is usually determined by deep-level defects. On Page 9 of the manuscript, we propose a qualitative physical picture that the trapping of carriers by iodine interstitials occurs in a fast time scale beyond our temporal resolution (see our Reply #1-2), whereas the slow detrapping of carriers back to the conduction/valence bands determines the measured lifetime. Further theoretical work is clearly needed to provide quantitative evidence for this possible scenario.

Changes #3-7: On Page 9 of the text, we already stated that the measured lifetime may be attributed to the trapping/de-trapping of carriers by iodine interstitials. We caution that this only provides a qualitative explanation and add a sentence to the revised manuscript “Further theoretical work is needed to elucidate this physical picture in a quantitative manner” on Page 9.

Review Comment #3-8: *Figure 2b: it is not clear for me if the data at each pixel are generated by time-integrated signals or at fixed time interval between optical excitation and microwave pulse. I'm confused since it will take time for carriers to diffuse outwards to reach the outside. That doesn't seem to be considered in the simple diffusion model, eqs. 1-3.*

Reply #3-8: This question is related to Reply #3-2 above. The diffusion mapping experiment in Fig. 1b (we assume that the Reviewer meant Fig. 1b rather than TRPL in Fig. 2b) is measuring the steady-state photoconductivity under CW laser excitation, where dynamic equilibrium is reached for various generation, recombination, and diffusion processes. The spatial distribution of free carriers under the equilibrium condition is described by the diffusion model Eqs. (1-3). There is no “time interval between optical excitation and microwave pulse” in this measurement.

Review Comment #3-9: *Some more minor questions that may deserve some further clarification: Is the excitation used close to 1 sun?*

Reply #3-9: We thank the Reviewer for this suggestion. As written in the manuscript, data shown in Figs. 1-3 were measured under an illumination intensity of 100 mW/cm². Since we used monochromatic laser excitation rather than white light in the experiment, it is more appropriate to denote the intensity as “on the order of 1 Sun”.

Changes #3-9: On Page 5 of the revised manuscript, we specify the intensity as “... $P_C = 100$ mW/cm², i.e., on the order of 1 Sun.”

Review Comment #3-10: *How do authors differentiate conductivity contrast vs topography contrast?*

Reply #3-10: In our previous work Ref. 33, we elaborated on the finite-element analysis of the topographic crosstalk. Specifically, we built 3D models that explicitly take into account the surface topography due to grains and grain boundaries. We then compared the experimental data and simulated results to identify the local conductivity at the GBs (Fig. 4 in Ref. 33). In this work, however, the emphasis is not on the deconvolution of electrical and topographic contributions. For instance, the diffusion analysis in Fig. 1e is averaged 8 line profiles, which

minimizes the topographic crosstalk. We believe that it is sufficient to cite Ref. 33 in the manuscript, as indicated on Page 5, and refer the readers to our detailed analysis there.

Review Comment #3-11: *Fig. S5: I expect different excitation energy will produce different diffusion length since hot carriers with different temperature will travel differently especially across grains.*

Reply #3-11: We agree with the Reviewer that hot carriers with different temperatures, if exist, should display different diffusion lengths. In our experiment, however, neither the lifetime nor the diffusion length depends on the laser wavelength. Our physical picture is as follow. Under above-gap illuminations, electron-hole pairs are excited to conduction/valence bands as hot carriers. Within a short timescale (ps or shorter), the carriers thermalize to the band edge by emitting phonons, after which the trapping/detrapping processes take place. The thermalization is very effective at low laser intensities (~ 1 Sun) such that the carrier temperature is essentially independent of the excitation wavelength. While it is possible to perform thorough power and wavelength-dependent diffusion studies, the effort would not make significant contribution to the current work. We believe that it is sufficient to briefly discuss the scenario in the SI section.

Changes #3-11: When discussing Fig. S5 in the Supplementary Information, we include the following discussions to address the Reviewer's comment – “It is possible that the thermalization to the band edge through phonon emitting is very effective at low laser intensities (~ 1 Sun), after which the trapping/detrapping processes take place. As a result, the carrier temperature is essentially independent of the excitation wavelength.”

Review Comment #3-12: *In summary, the manuscript contains interesting data using a powerful spatial-temporal imaging technique. It will be much more informative if the above questions can be addressed.*

Reply #3-12: We thank the Reviewer for his/her highly positive comments on our technique and the quality of our data. We believe the above questions are fully resolved by our answers.

To summarize, we are very grateful to receive the constructive reviews from all three referees. With all their comments addressed, we hope that the manuscript is now suitable for publication in *Nature Communications*. Thank you very much for your consideration.

Best regards,

Keji Lai
Associate Professor of Physics, University of Texas at Austin
Austin, TX 78712 USA

REVIEWER COMMENTS

Reviewer #1 (Remarks to the Author):

Although I still hold concerns about the using of terminology of 'dynamics' in a method mainly reflected the photo-physics at equilibrium condition, I think the explanation from the authors can be accepted. However, it would be better to provide detailed statements on the speciality of the iMIM measurement with what has been explained in the response letter. By this means, the readers would know better the expertise as well as the limitation of the technique for the characterization on PV devices.

Reviewer #3 (Remarks to the Author):

I have read the revision carefully and I appreciate the detailed responses that authors provide to my and other referees' questions. My overall impression is below: the measurement scheme is new and interesting but the physical insights obtained is still compromised and may not make a clear cut to Nature Communications. The major drawback is that the time resolution is not enough to truly resolved non-equilibrium charge transfer but rather a quasi-equilibrium states of at least 100ns that is only relevant to some of the processes at long times such as trapping and de-trapping. The excitations is not local and cannot selective address individual grain and grain boundaries. Rather the signals are still an average of charge transfer of many grain and boundaries. Although the tr-MIM is still a powerful way that are complementary to other spatially- and temporally-resolved techniques using pump-probe and PL, the new physics discovered in this paper so far are not novel enough in the current format. It may need some extra efforts to improve the novelty which I elaborate below:

1. The main conclusions include: (1) the inhomogenous distribution of the photoconductivity over the film structure; (2) the different relaxation times and mobility of electrons and holes. The first one has been well established in the perovskite community. The second one is more interesting but as far as I know they have also been discussed in various context in recent literature. Does the iMIM results here the first one to selectively measure diffusion length and lifetimes of electrons and holes? I'm not so sure about them. If yes, the authors should emphasize this and provide a more rigorous physical picture on the disparity of their mobility and lifetimes that substantiate the observation. This will improve the novelty significantly in my opinion.
2. It is still highly surprising to me why the lifetimes have no difference at all between grains and grain boundaries. There is no reason one cannot see any difference with 20 nm and 1 ns resolutions. It has been well established that they have different defect densities and build-in potentials among many other key transport parameters. It may be superficial to still think that grain boundaries are benign to charge transfer and dynamics. I don't have doubt on the iMIM data in Fig. 2d but they may only provide a partial measure of the whole processes which can have a misleading view due the resolutions. Anyway, those should be carefully articulated to avoid any confusions to add into the current perovskite literature that is filled with many inaccurate claims. At least authors should provide a reasonable discussion on the physical picture behind their claims instead of only presenting the observation.
3. I am still confused with the MIM contrast - grain boundary should have very limited mobility in comparison to grains. Why do they have bright contrast in Fig. 2C? Does that originate from strong carrier trapping near the grain boundaries? However the trapped carriers are not mobile and how that affects the MIM contrast. In addition, some of the grains in Fig. 2C are very dark – does that mean there are very few carries in those grains in comparison to the boundaries? Can mobility difference between grains and boundaries account some of the difference? Some of the grains in Fig. 2C are very bright also. What happen there? I'm still confused by these contrasts in the revised manuscript.

Reply to Reviewer 1's report:

Review Comment #1-1: *Although I still hold concerns about the using of terminology of 'dynamics' in a method mainly reflected the photo-physics at equilibrium condition, I think the explanation from the authors can be accepted. However, it would be better to provide detailed statements on the specialty of the iMIM measurement with what has been explained in the response letter. By this means, the readers would know better the expertise as well as the limitation of the technique for the characterization on PV devices.*

Reply #1-1: We thank the Reviewer for accepting our explanation in the previous response letter. In terms of the terminology, the tr-iMIM is detecting the dynamic response of photo-excited carriers. The diffusion mapping also measures the dynamic equilibrium under continuous carrier generation and recombination. Nevertheless, we agree with the Reviewer that it is important to provide detailed statements on the specialty of iMIM, which were thoroughly explained in our previous response letter, in the revised manuscript. Specifically, we will add statements on (1) what steady-state iMIM is measuring and how it is related to PV devices; (2) what tr-iMIM is measuring and how it is related to PV devices; (3) the limitation of the technique by a comparison with PL.

Changes #1-1: On Page 4 of the revised manuscript, we highlight the specialty of the iMIM measurement as “The steady-state iMIM experiment addresses the most important photophysical process in solar cells, i.e., the transport of photo-generated mobile carriers under the continuous illumination of ~ 1 Sun. The time-resolved iMIM (tr-iMIM) experiment detects the free-carrier lifetime that is also highly relevant for photoconduction.” On Page 9 of the paper, we also include a statement to compare PL and iMIM measurements as “In short, PL measured excited states such as exciton recombination via emitted photons, whereas iMIM measures near-equilibrium conductivity following optical injection of free carriers.” We hope that the concerns from Reviewer 1 have now been fully resolved.

Reply to Reviewer 3's report:

Review Comment #3-1: *I have read the revision carefully and I appreciate the detailed responses that authors provide to my and other referees' questions. My overall impression is below: the measurement scheme is new and interesting, but the physical insights obtained is still*

compromised and may not make a clear cut to Nature Communications. The major drawback is that the time resolution is not enough to truly resolved non-equilibrium charge transfer but rather a quasi-equilibrium states of at least 100ns that is only relevant to some of the processes at long times such as trapping and detrapping. The excitation is not local and cannot selectively address individual grain and grain boundaries. Rather the signals are still an average of charge transfer of many grains and boundaries. Although the tr-MIM is still a powerful way that are complementary to other spatially- and temporally-resolved techniques using pump-probe and PL, the new physics discovered in this paper so far are not novel enough in the current format. It may need some extra efforts to improve the novelty which I elaborate below:

Reply #3-1: We thank the Reviewer for pointing out the new and interesting measurement scheme here. Our general response to his/her comments is as follows. Regarding the time resolution, the tr-iMIM is a truly non-equilibrium measurement on free-carrier lifetimes, which are determined by relatively slow (e.g., carrier detrapping from defects) processes in hybrid perovskites. Since the lifetime is proportional to the steady-state carrier density under solar illumination, it is one of the most important time scales for photovoltaic applications. Regarding the non-local (1~2 μm) excitation, we agree with the Reviewer that it is difficult to selectively address individual grains and grain boundaries (GBs), which will be elaborated in Reply #3-3 below. Nevertheless, the local (100nm) detection enables us to accurately fit the spatial distribution of photoconductivity based on the diffusion equation and laser profile, from which the diffusion length across many grains and GBs is extracted. In short, the iMIM indeed complements pump-probe and TRPL experiments to provide crucial information on solar cell studies. We are now ready to improve the novelty statement in our manuscript.

Review Comment #3-2: *The main conclusions include: (1) the inhomogeneous distribution of the photoconductivity over the film structure; (2) the different relaxation times and mobility of electrons and holes. The first one has been well established in the perovskite community. The second one is more interesting but as far as I know they have also been discussed in various context in recent literature. Does the iMIM results here the first one to selectively measure diffusion length and lifetimes of electrons and holes? I'm not so sure about them. If yes, the authors should emphasize this and provide a more rigorous physical picture on the disparity of their mobility and lifetimes that substantiate the observation. This will improve the novelty significantly in my opinion.*

Reply #3-2: We are grateful to the Reviewer for this insightful suggestion. First of all, it is never our intent to highlight the inhomogeneous photoconductivity over grains and GBs, which is not even mentioned in the Abstract. On the other hand, we completely agree with him/her that we should emphasize the quantification of relaxation time and mobility of electrons and holes, which is indeed our main focus. Prior to our work, separate experiments are needed to measure relaxation time (TRPL or TRTS) and carrier mobility (transport on FETs or SPCM on doped samples). Moreover, it is not trivial to individually address the lifetime/mobility of free electrons and holes. For instance, in TRPL measurements, the radiative recombination is quenched when one type of carriers is extracted by HTL or ETL, and the decay constants no longer represent electron/hole lifetimes in plain perovskite films. Similarly, in TRTS experiments (Ref. 17 and Fig. S9), the two measured mobilities cannot be unambiguously associated with electrons or holes. To our knowledge, it is the first time that diffusion length, carrier lifetime, and mobility can be individually addressed for mobile electrons and holes on the same batch (as-grown, HTL-

coated, ETL-coated) of samples, which should be stressed in the manuscript. Finally, we do have one paragraph on Pages 9-10 to speculate on the origin of disparate lifetime/mobility of electrons and holes. Some of us (X.W. and Y.Y.) are currently conducting theoretical analysis to understand the effect of iodine interstitials. Since that effort is clearly beyond the scope of the present work, our qualitative discussion is sufficient to provide a physical picture to the readers.

Changes #3-2: The key points enumerated above are repeatedly addressed throughout the manuscript. In order to substantiate our observation, we add the statements to the Conclusion part – “We emphasize that, prior to our work, separate experiments are needed to measure relaxation time (TRPL or TRTS) and mobility (transport or SPCM on doped samples) of free carriers. To our knowledge, it is the first time that diffusion length, carrier lifetime, and charge mobility can be individually addressed for mobile electrons and holes on the same batch (as-grown, HTL-coated, ETL-coated) of samples.” We believe that the novelty of this work is now sound and clear.

Review Comment #3-3: *It is still highly surprising to me why the lifetimes have no difference at all between grains and grain boundaries. There is no reason one cannot see any difference with 20 nm and 1 ns resolutions. It has been well established that they have different defect densities and build-in potentials among many other key transport parameters. It may be superficial to still think that grain boundaries are benign to charge transfer and dynamics. I don't have doubt on the iMIM data in Fig. 2d but they may only provide a partial measure of the whole processes which can have a misleading view due the resolutions. Anyway, those should be carefully articulated to avoid any confusions to add into the current perovskite literature that is filled with many inaccurate claims. At least authors should provide a reasonable discussion on the physical picture behind their claims instead of only presenting the observation.*

Reply #3-3: We appreciate this critical comment from the Reviewer. Indeed, the nonlocal photo-excitation (diffraction-limited to 1~2 μm) does not allow us to selectively address individual grains and GBs in this particular sample (grain size < 1 μm). As a result, from the iMIM data in Fig. 2d, one can only conclude that the measured τ_1 , τ_2 , and A_1/A_2 are uniform across the sample surface within statistical errors. Going beyond this point, as indicated by the Reviewer, may “add into the current perovskite literature that is filled with many inaccurate claims”. Since it is not our intent to resolve the debate on the nature of GBs in this work, we decide to move Figs. 2c and 2d and the corresponding texts to Supplementary Information S6, with clear description of the results and discussion on the limitation. This change does not alter the flow of our presentation or the main conclusion of the paper.

With that said, we hold firm ground on the observation of carrier diffusion across multiple grains and GBs in our samples, as indicated by the photoconductivity maps (Fig. 1b and Fig. 3b/3c). We understand the Reviewer's concern regarding other reports showing various degrees of defect densities and band bending at GBs that can affect transport parameters. It is possible that the GB properties depend on the perovskite compositions and processing conditions, which may vary in different studies. Our data suggest that the GBs *in the current study* are not strong nonradiative recombination (i.e., highly defective) centers, and there is no significant band bending at the GBs to block electron/hole movement across multiple grains. In the future, we plan to use the iMIM, possibly together with PL techniques, to examine other state-of-the-art perovskite materials. As for the current study, we believe that a brief mention of such possibility is sufficient.

Changes #3-3: Major revisions are made in the manuscript to address this comment:

- 1) On Page 4 of the revised paper, we remove the statement “... ~~which are uniform across grain and grain boundaries, ...~~”
- 2) On Page 6, we remove the description of Figs. 2c and 2d “~~The spatial uniformity of the tr-iMIM response was investigated as follows. We first took the AFM/iMIM images (Supplementary Fig. S6) in the sample scan mode under CW laser with an ultra sharp tip (Fig. 2c). We then parked the tip on various points on top of grains and GBs for tr-iMIM measurements (Supplementary Fig. S7). As plotted in Fig. 2d, there is no difference between grains and GBs in terms of τ_1 , τ_2 , and A_1/A_2 within statistical errors. We, therefore, conclude that the temporal dynamics of photo carriers are spatially uniform across microstructures on the sample surface.~~”
- 3) The point measurements are briefly discussed on Page 6 as “~~By parking the tip at various locations of the film and measuring the decay curves, we also show that the tr-iMIM response is spatially uniform on the sample surface within statistical errors (Supplementary Figs. S6 and S7).~~”
- 4) On Page 8, we remove the previous statement “~~Moreover, the temporal evolution of photoconductivity does not differ between grains and GBs within statistical uncertainties.~~”
- 5) In order to explain the carrier diffusion across multiple grains and GBs in our samples, we provide a hypothesis on Page 8 as “~~It is possible that the GBs in the current study are not strong nonradiative recombination (i.e., highly defective) centers, and there is no significant band bending at the GBs to block electron/hole movement across multiple grains⁴⁸.~~”
- 6) On Page 10, we remove “...~~uniformly across grains and GBs...~~”
- 7) A new reference (Ref. 48, Lin, Y. et al. “Perovskite solar cells with embedded homojunction via nonuniform metal ion doping”, *Cell Rep. Phys. Sci.* **2**, 100415 (2021).) is cited to note the possible band bending at the GBs.
- 8) Panels 2c and 2d, as well as the corresponding text, are removed from Fig. 2
- 9) The point measurements (previous Fig. 2d) are moved to Fig. S6d. The corresponding explanation reads as “~~We then park the tip at various points on top of grains and GBs for tr-iMIM measurements, as labeled in Fig. S6b. As plotted in Fig. S6d, the measured τ_1 , τ_2 , and A_1/A_2 are spatially uniform across the sample surface within statistical errors. We caution that, since the laser spot ($\sim 2 \mu\text{m}$) is larger than the average grain size here, the result does not guarantee the conclusion that the temporal dynamics of photo-carriers is the same for grains and GBs. Further experiments with smaller laser spots and large-grain samples are needed for this investigation.~~”

Review Comment #3-4: *I am still confused with the MIM contrast - grain boundary should have very limited mobility in comparison to grains. Why do they have bright contrast in Fig. 2C? Does that originate from strong carrier trapping near the grain boundaries? However the trapped carriers are not mobile and how that affects the MIM contrast. In addition, some of the grains in Fig. 2C are very dark – does that mean there are very few carries in those grains in comparison to the boundaries? Can mobility difference between grains and boundaries account some of the difference? Some of the grains in Fig. 2C are very bright also. What happen there? I'm still confused by these contrasts in the revised manuscript.*

Reply #3-4: The iMIM contrast in the sample-scan mode (previously Fig. 2c, now Fig. S6b) is mostly due to topographic crosstalk, a common issue for near-field microscopy. In this mode of operation, the tip is first aligned to the center of the laser spot, followed by raster scan of the sample. When tracking the rough sample surface, the tip “senses” more surrounding materials as it dips into the trench-like GBs, and less surrounding materials as it climbs up to tall grains. As a result, the contrast between microstructures in this mode is convoluted with topography and does not carry significant information. In our earlier work (Ref. 33 or Ref. S3), we scratched away part of the active layer such that the iMIM signal could be compared with that of the substrate. In this work, we are focusing on the tip-scan mode, where the tip moves to regions with no illumination for comparison. Since we have now moved Figs. 2c and 2d to the Supplementary Information, we do not wish to spend much effort to explain the convoluted contrast mechanism in the main text. A brief description on topographic crosstalk is provided in the text of Fig. S6. We also refer the interested readers to Ref. 33 or Ref. S3 for more discussions.

Changes #3-4: In Supplementary Information Section S6, we provide the following description of the topographic crosstalk – “Sample-scan AFM and iMIM-Im/Re images on Sample A are shown in Fig. S6. The tip is first aligned to the center of the laser spot, followed by raster scan of the sample. The contrast between grains and grain boundaries (GBs) in the iMIM images is mostly due to topographic crosstalk. In short, when tracking the rough sample surface, the tip “senses” more surrounding materials as it dips into the trench-like GBs, and less surrounding materials as it climbs up to tall grains. As a result, the contrast between microstructures in this mode is convoluted with topography, which is a common issue for near-field microscopy^{S3}”.

To summarize, we are very grateful to receive the constructive reviews from both referees. With all their comments addressed, we hope that the manuscript is now suitable for publication in *Nature Communications*. Thank you very much for your consideration.

Best regards,

Keji Lai
Associate Professor of Physics, University of Texas at Austin
Austin, TX 78712 USA

REVIEWERS' COMMENTS

Reviewer #3 (Remarks to the Author):

I'm generally happy with authors' answers to my comments. They also agree that there can be artifacts due to nonlocal excitations and the origin of disparate lifetime/mobility of electrons and holes can only be speculated at the moment. I appreciate authors' honest statements and clarifications on those issues. Although I still have an issue to use "non-equilibrium measurement" to describe tr-iMIM since it is only sensitive to the very slow process such as carrier detrapping from defects and neglect almost all the initial and intermediate transient processes. I agree though tr-iMIM indeed complements pump-probe and TRPL experiments to provide an interesting, alternative view. The authors have also done extensive work to clean up the flow and articulate the main novelties and conclusions. I would suggest authors to avoid use "non-equilibrium measurement" but rather say it is a time-resolved measurement of microsecond lifetimes. With that I can recommend the paper for publication.

Reply to Reviewer 3's report:

Review Comment:

I'm generally happy with authors' answers to my comments. They also agree that there can be artifacts due to nonlocal excitations and the origin of disparate lifetime/mobility of electrons and holes can only be speculated at the moment. I appreciate authors' honest statements and clarifications on those issues. Although I still have an issue to use "non-equilibrium measurement" to describe tr-iMIM since it is only sensitive to the very slow process such as carrier detrapping from defects and neglect almost all the initial and intermediate transient processes. I agree though tr-iMIM indeed complements pump-probe and TRPL experiments to provide an interesting, alternative view. The authors have also done extensive work to clean up the flow and articulate the main novelties and conclusions. I would suggest authors to avoid use "non-equilibrium measurement" but rather say it is a time-resolved measurement of microsecond lifetimes. With that I can recommend the paper for publication.

Reply: We thank the Reviewer for the constructive reviews and recommending the paper for publication after revision. We understand the concern about using the word "non-equilibrium" in our previous reply letter. Thus, we modify the description in the manuscript accordingly as below.

Changes: On Page 3 and Page 8 of the revised manuscript, we modified the sentences to "In contrast, the transport of photo-excited carriers is electrical and **quasi-static** in nature" and "TRPL measured excited states such as exciton recombination via emitted photons, whereas **tr-iMIM** measures **the decay of steady-state** conductivity following optical injection of free carriers."

To summarize, we are very grateful to receive the constructive review from the referee. With his/her final comment addressed, we hope that the manuscript is now suitable for publication in *Nature Communications*. Thank you very much for your consideration.

Best regards,

Keji Lai
Associate Professor of Physics, University of Texas at Austin
Austin, TX 78712 USA